# Evaluation of ESV Change under Urban Expansion Based on Ecological Sensitivity: A Case Study of Three Gorges Reservoir Area in China

Hongjie Peng [1,2], Lei Hua [3], Xuesong Zhang [1,2,*], Xuying Yuan [1,2] and Jianhao Li [1,2]

[1] Department of Geographic Information Science, College of Urban and Environmental Sciences, Central China Normal University, Wuhan 430079, China; penghongjie@mails.ccnu.edu.cn (H.P.); Yuanxuying@mail.ccnu.edu.cn (X.Y.); ljh@mails.ccnu.edu.cn (J.L.)

[2] Key Laboratory for Geographical Process Analysis & Simulation, Wuhan 430079, China

[3] Department of Public Service Management, School of Public Affairs, Chongqing University, Chongqing 400044, China; hualei@mails.ccnu.edu.cn

* Correspondence: zhangxuesong@mail.ccnu.edu.cn

**Abstract:** In recent years, ecosystem service values (ESV) have attracted much attention. However, studies that use ecological sensitivity methods as a basis for predicting future urban expansion and thus analyzing spatial-temporal change of ESV are scarce in the region. In this study, we used the CA-Markov model to predict the 2030 urban expansion under ecological sensitivity in the Three Gorges Reservoir area based on multi-source data, estimations of ESV from 2000 to 2018 and predictions of ESV losses from 2018 to 2030. Research results: (i) In the concept of green development, the ecological sensitive zone has been identified in Three Gorges Reservoir area; it accounts for about 35.86% of the study area. (ii) It is predicted that the 2030 urban land will reach 211,412.51 ha by overlaying the ecological sensitive zone. (iii) The total ESV of Three Gorges Reservoir area showed an increasing trend from 2000 to 2018 with growth values of about USD 3644.26 million, but the ESVs of 16 districts were decreasing, with Dadukou and Jiangbei having the highest reductions. (iv) New urban land increases by 80,026.02 ha from 2018 to 2030. The overall ESV losses are about USD 268.75 million. Jiulongpo, Banan and Shapingba had the highest ESV losses.

**Keywords:** ecological sensitivity; ecosystem service values; CA-Markov model; urban expansion; Three Gorges Reservoir area

## 1. Introduction

An ecosystem is a unity of the biota and the abiotic surroundings in a certain space of nature [1]. It is also an ecological functional unit formed by the continuous exchange of energy and materials between organisms and their abiotic environment [2–4]. Ecosystems can provide a range of services for human production and livelihoods [5–7]. Ecosystem services are the materials that humans need to obtain from the Earth's ecosystems and natural environment [8,9]. Ecosystem services are diverse, and the services are unique and irreplaceable for different ecosystem [10–12]. For example, forests are the most essential ecosystems for human life, providing wood, regulating air, promoting soil formation and supporting social benefits such as spiritual, landscape and educational values [13]. The evaluation of ESV can measure the status of a regional ecosystem.

The evaluation of ESV is a method based on ecology, economics and sociology [14–16]. It can quantitatively evaluate ecosystem services from the perspective of monetary value and better reflect the change in ESV and provide decision makers with knowledge to make policy adjustments [16,17]. In the early days, its evaluation methods made some progress in emphasizing the impact of socio-environmental change on ecosystem services (supply, regulation, support and entertainment services) [18–20]. Research scholars began to use

RS and GIS technology to assess ESV for the region by remote sensing technology, such as China at the national scale [9,21] and Leipzig at the municipal scale [22].

The delimitation of the ecological sensitive zone is important to improve ESV in the region [5]. It is believed that an ecological sensitive zone is characterized by the low stability of an ecosystem that is easily affected by external activities, ecological degradation and difficulty with self-restoration [23]. When humans develop them unreasonably, ecological sensitive zones are prone to environmental problems. Delineating the ecological sensitive zone plays an important role in maintaining the health of the ecosystem [24,25]. The ecological sensitivity evaluation method can quantitatively identify the ecological sensitive zone. The evaluation results provide the possibility to effectively control and protect the target zone [7]. It also provides a practical way for developing countries to realize its regional sustainable development strategy [26]. Starting from the perspective of ecological sensitivity, providing scientific guidance methods for realizing the sustainable development of regional ecosystem [27].

The assessment of ESV is closely linked to land use patterns. Land use simulations can be modelled using spatial tools such as cellular automata (CA), which are temporal, spatial and state-discrete models that can simulate complex dynamic systems with spatial characteristics [28–31]. CA model express geographic entity information to simulate and predict complex geographic processes by constructing a systemic spatial concept system [32]. Thus, CA is commonly used to predict future land use changes through endogenous and exogenous drivers, which have the roles of human disturbance as well as socioeconomic and institutional factors [33,34]. Estimations of future ESVs are obtained by addressing the interactions between drivers, land use changes and ESV changes. These interactions are usually considered as the dependent variable component of the CA model. For example, logistic regression is combined with a CA model, which is used by assuming that the development probability of a location is a function of a set of independent variables [35]. The ANN-CA model is a combination of artificial neural networks (ANN) and CA models, which combines the nonlinear processing capability of artificial neural networks and the spatial simulation capability of CA models [36]. In the CA-Markov model, the Markov chain process controls the temporal variation between land use types according to the transition matrix, which facilitates the use of a wider range of spatial variables and improves the accuracy of the model [37,38].

As urbanization progresses, it is necessary to delimit ecological sensitive zones and predict the future urban development direction of the region through the CA-Markov model to promote sustainable urbanization development. On this basis, estimating the dynamic change of ESV in the region will provide a full understanding of the services provided by ecosystems for human social development, providing relevant reference for ecological environmental protection and the comprehensive control of ecological reserves. Therefore, this paper takes the Three Gorges Reservoir area as an example to simulate the future development rate and direction of urban land in different districts and counties of the Three Gorges Reservoir area under the role of the ecological sensitive zone and urban land drivers. The quantitative estimation of change in ESV and the prediction of future ESV losses in the region are performed by ecosystem service value equivalence tables. This study can provide scientific reference to promote ecological civilization construction and sustainable development in developing countries.

## 2. Materials and Methods

### 2.1. Description of the Study Area

The Three Gorges Reservoir area is located at the end of the upper reaches of the Yangtze River Basin, with an area of approximately 68,772.93 km$^2$ (Figure 1). It is bordered by three provinces and a municipality, namely Hubei Province, Sichuan Province, Guizhou Province and Chongqing City. The Three Gorges Reservoir area includes 26 districts and counties (autonomous regions). The relative height shows a gradual increase from west to

east. The central and western parts of the reservoir area are dominated by platforms and hills. The eastern part is close to the Daba Mountain and has many rolling hills [39].

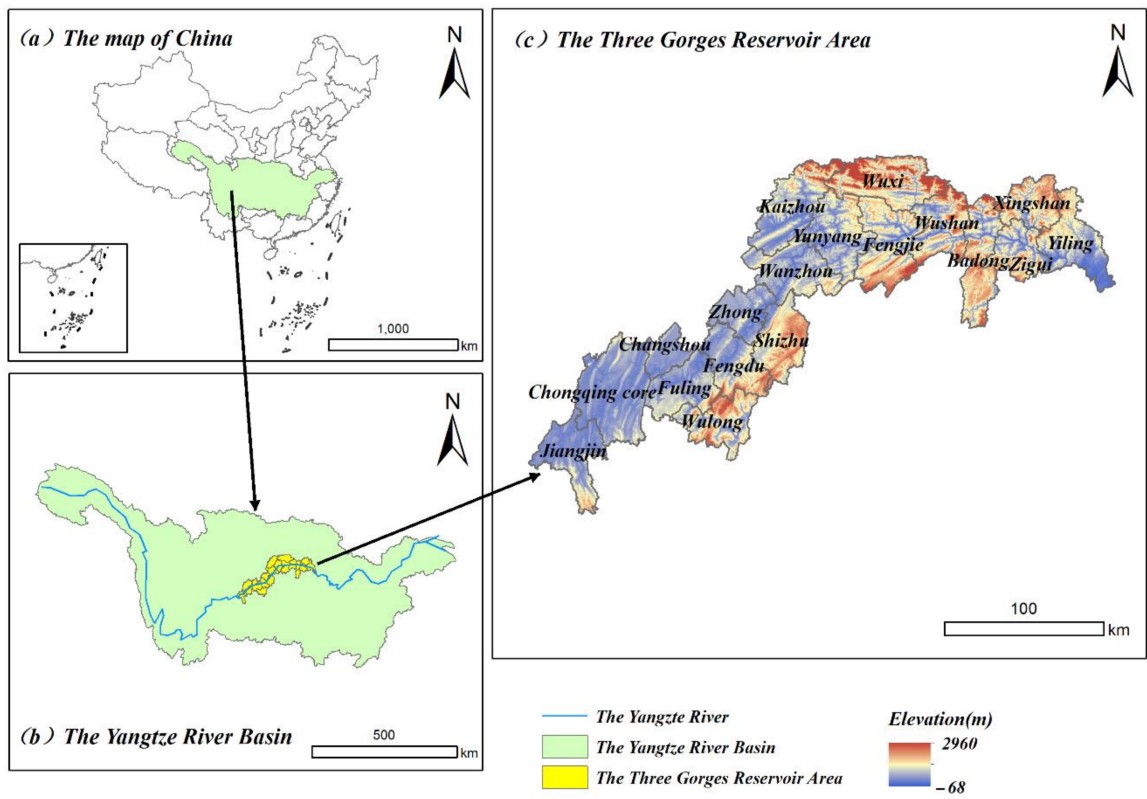

**Figure 1.** Location and elevation of the study area. (**a**) The map of China; (**b**) the Yangtze River basin; (**c**) the Three Gorges Reservoir area.

There are abundant vegetation types in the Three Gorges Reservoir area, and the overall spatial distribution of vegetation coverage shows the characteristics of high in the east and low in the west [40]. The east is mostly broad-leaved forests, bushes and grasslands; the central and western area are farming area with many cultivated plants and crops [41]. The Three Gorges Reservoir area is an important ecological barrier in the Yangtze River Basin, China's strategic water resources reserve. Therefore, the prediction of ESV change in the Three Gorges Reservoir area is important for China's sustainable development under the future urban expansion.

*2.2. Materials*

The two main types of data used in this study are spatial data and statistical yearbook data.

Spatial Data. (1) Administrative boundary vector data of Three Gorges Reservoir area (SHP format). (2) Soil dataset provided by Harmonized World Soil Database (HWSD) and Cold Arid Regions, Available online: http://www.westdc.westgis.ac.cn (accessed on 17 April 2019), which contains the spatial coordinates and properties of the soil (GRID format). (3) Digital elevation model (DEM), which was downloaded from the Geospatial Data Cloud Available online: http://www.gscloud.cn/ (accessed on 26 April 2020). In the above data, the DEM data, with a resolution of 30 m, can be extracted into slope and elevation. (4) The highway, the primary road and railroad data procured from OpenStreetMap Available online: http://www.openstreetmap.org (accessed on 13 July 2020). The population density data was provided by Landscan Available online: https://landscan.ornl.gov/ (accessed on 18 December 2020). (5) The 1000 m Normalized Difference Vegetation Index (NDVI) data and 30 m land use and land cover change (LUCC) data were obtained from the Data

Centre for Resources and Environmental Sciences, Chinese Academy of Sciences (RESDC) Available online: http://www.resdc.cn (accessed on 26 April 2020). NDVI and LUCC data were obtained by remote sensing image processing. NDVI data was based on inversion of SPOT/VEGETATION and MODIS satellite remote sensing and LUCC data was generated by manual visual interpretation based on Landsat 8 remote sensing images. (6) China Nature Reserve, the earthquake and landslide vector data acquired from RESDC. China Nature Reserve are surface vector data, the rest are point vector data.

Statistical data: The statistical data include average annual rainfall (2018), average annual temperature (2018), number of days with wind and sand wind speed ≥10 m/s (2018), annual evaporation (2018), groundwater mineralization and groundwater burial depth. Meteorological data were mainly obtained from 22 meteorological stations around the Three Gorges Reservoir area by the Chinese Meteorological Science Data Sharing Service Available online: http://data.cma.cn/site/index.html (accessed on 17 January 2020). According to each weather station, the inverse distance interpolation (IDW) method of ArcGIS was used to convert meteorological statistics into a raster image of meteorological data in the study area. Groundwater mineralization and groundwater depth of burial data were obtained by querying the statistical yearbooks of different districts and counties in the Three Gorges Reservoir area.

### 2.3. Methods

In this paper, we use the overlay analysis function of GIS, Markov chain and cellular automata (CA) to simulate the future urban expansion. From the scenario of ecological conservation, predicting future urban expansion, estimating ESV during 2018–2030 and forecasting ESV losses at 2030 in the Three Gorges Reservoir area were performed. The research framework of this study is presented in Figure 2.

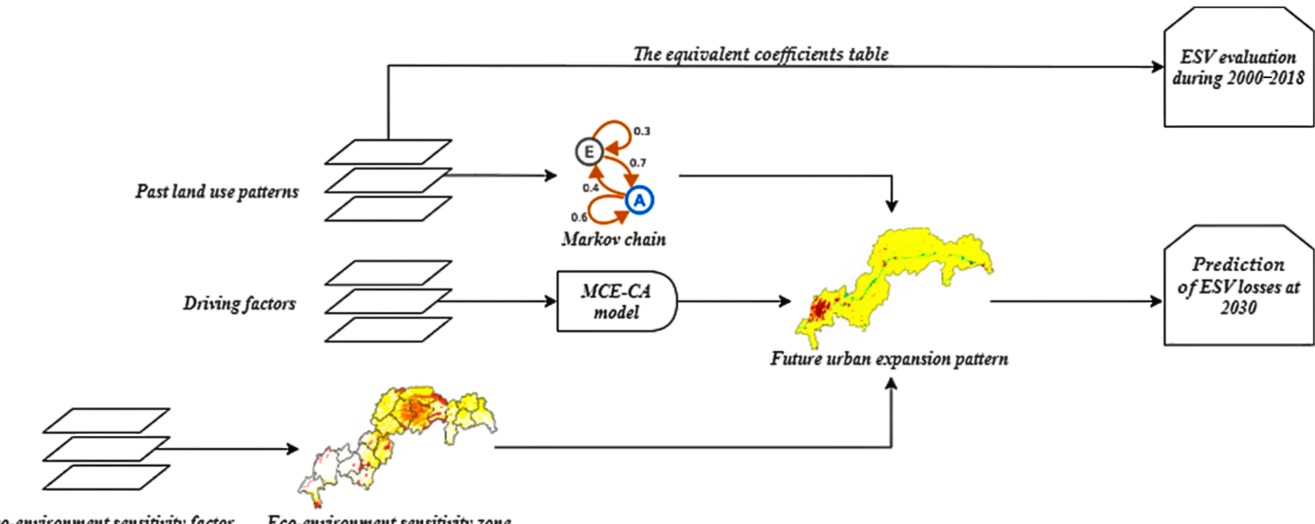

**Figure 2.** Research framework.

### 2.3.1. Ecological Comprehensive Sensitivity and Zone Identification

In this research, the comprehensive ecological sensitivity was calculated from three sensitivities of soil erosion, land desertification and soil salinization. The calculation formula is shown below:

$$EES = \omega a * SS_i + \omega b * Z_i + \omega c * S_i \tag{1}$$

where EES denotes the comprehensive ecological sensitivity, and $SS_i$, $Z_i$ and $S_i$ are the weight values of soil erosion, land desertification and soil salinization, respectively. The

coefficient of variation method was used to calculate $\omega_a$, $\omega_b$ and $\omega_c$ as 49.63%, 32.00% and 18.37%, respectively [42].

$R_i$, $SG_i$, $LS_i$ and $C_i$ were selected for the evaluation of $SS_i$. The ecological sensitivity factors were classified into five classes named insensitive, mildly sensitive, moderately sensitive, highly sensitive and extremely sensitive, which were categorized into classes 1, 2, 3, 4 and 5, respectively (Table 1). The four indicators are calculated as follows:

$$SSi = \sqrt[4]{Ri * SGi * LSi * Ci} \tag{2}$$

where $SS_i$ denotes the soil erosion sensitivity; $R_i$ is the rainfall erosivity factor; $SG_i$ is the soil type factor; $LS_i$ is the relative height factor; and $C_i$ is the normalized difference vegetation index factor. The ecological sensitivity factors were divided into five classes named insensitive, mildly sensitive, moderately sensitive, highly sensitive and extremely sensitive, which were categorized into classes 1, 2, 3, 4 and 5 (Table 1).

**Table 1.** Criteria for the soil erosion sensitivity.

| Sensitivity Degree | $R_i$ | $SG_i$ | $LS_i$ | $C_i$ |
|---|---|---|---|---|
| Insensitive (1) | <25 | Paddy soil, urban area, rock and river | <20 | >0.49 |
| Mildly sensitive (2) | 25–100 | Limestone soil, rock-soil and mountain meadow soil | 20–50 | 0.39–0.49 |
| Moderately sensitive (3) | 100–400 | Dark brown soil and yellow-cinnamon soil | 50–100 | 0.28–0.39 |
| Highly sensitive (4) | 400–600 | Yellow loam, yellow-brown soil and skeleton soil | 100–300 | 0.16–0.28 |
| Extremely sensitive (5) | >600 | Purple soil | >300 | <0.16 |

The evaluation of the $Z_i$ required the $I_i$, $W_i$ and $SL_i$. Its classification and assignment methods were the same as those of $SS_i$ (Table 2). The formula of $Z_i$ was as follows:

$$Zi = \sqrt[3]{Ii * Wi * SLi} \tag{3}$$

where $Z_i$ denotes the land desertification sensitivity; $I_i$ is the dryness index factor; $W_i$ is the number of days on which wind-blown sand speeds are 6m/s; $SL_i$ is the slope factor.

**Table 2.** Criteria for the land desertification sensitivity.

| Sensitivity Degree | $I_i$ | $W_i$ | $SL_i$ |
|---|---|---|---|
| Insensitive (1) | <0.96 | <2 | ≤5 |
| Mildly sensitive (2) | 0.96–1.01 | 2–4 | 5–8 |
| Moderately sensitive (3) | 1.01–1.08 | 4–6 | 8–15 |
| Highly sensitive (4) | 1.08–1.17 | 6–8 | 15–25 |
| Extremely sensitive (5) | >1.17 | >8 | >25 |

The evaluation of the $S_i$ required the $E_i$, $GS_i$, $GD_i$ and $LUCC_i$. Its classification and assignment methods were the same as those of $SS_i$ (Table 3). The formula of $S_i$ is as follows:

$$Si = \sqrt[4]{Ei * GSi * GDi * LUCCi} \tag{4}$$

where $S_i$ is the salinization sensitivity; $E_i$ is the evaporation index factor; $GS_i$ is the degree of mineralization of ground water factor; $GD_i$ is the groundwater depth factor; $LUCC_i$ is the land use and land cover change factor.

**Table 3.** Criteria for the salinization sensitivity.

| Sensitivity Degree | $E_i$ | $GS_i$ | $GD_i$ | $LUCC_i$ |
|---|---|---|---|---|
| Insensitive (1) | <0.6 | <0.1 | >20 | Water body |
| Mildly sensitive (2) | 0.6–0.7 | 0.1–0.2 | 18–20 | Impervious surface |
| Moderately sensitive (3) | 0.7–0.8 | 0.2–0.3 | 16–18 | Forest and grassland |
| Highly sensitive (4) | 0.8–0.9 | 0.3–0.4 | 14–16 | Farmland |
| Extremely sensitive (5) | >0.9 | >0.4 | <14 | Unused land |

The assessment results of ecological sensitivity were divided into 5 degrees, namely, insensitive, mildly sensitive, moderately sensitive, highly sensitive and extremely sensitive. The eleven factors required to calculate the spatial distributions of $SS_i$, $Z_i$ and $S_i$ were presented in Figure 3.

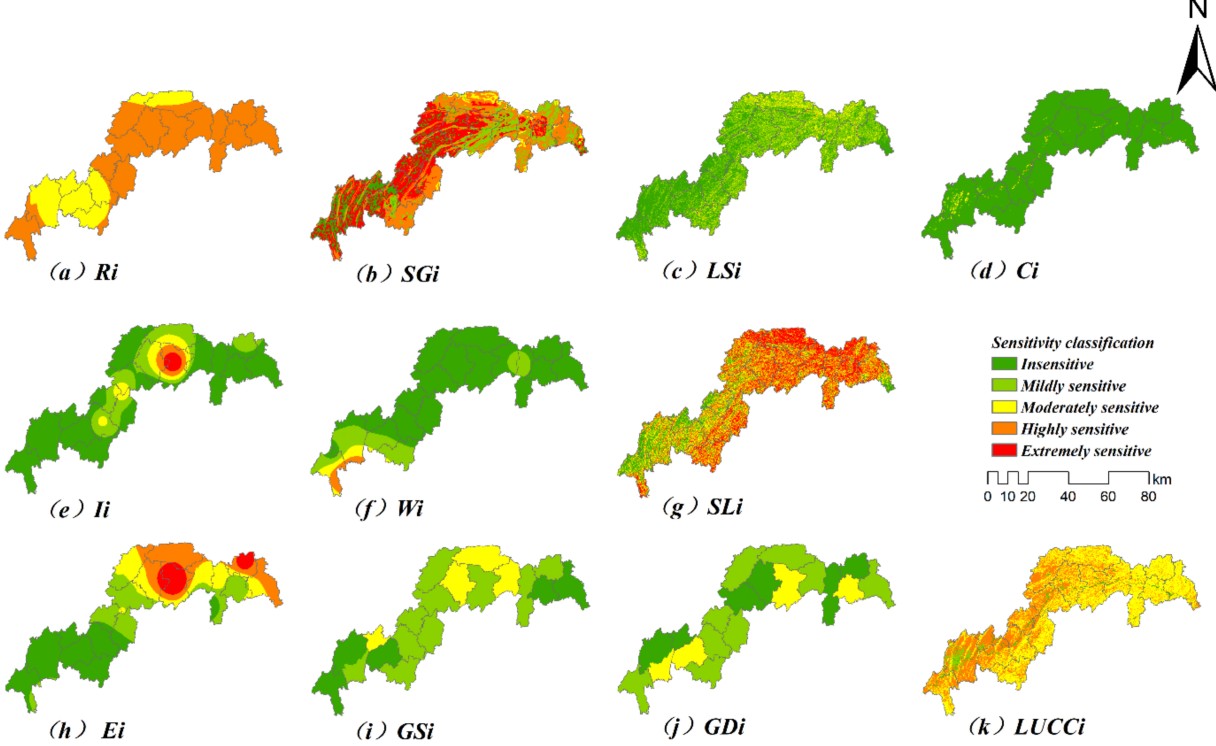

**Figure 3.** The spatial distribution of ecological sensitivity of 11 indicators in 2018: (**a**) rainfall erosion sensitivity; (**b**) soil type sensitivity; (**c**) relief of topography sensitivity; (**d**) vegetation cover sensitivity; (**e**) dryness sensitivity; (**f**) wind-blown sand sensitivity; (**g**) slope sensitivity; (**h**) evaporation sensitivity; (**i**) mineralization of ground water sensitivity; (**j**) groundwater depth sensitivity; (**k**) land use and land cover change sensitivity.

Based on the results of the $SS_i$, $Z_i$, $S_i$ and EES (Figure 4), using the natural breakpoint method in ArcGIS, the high sensitivity zone in the EES is proposed to determine the location of ecological sensitive zone.

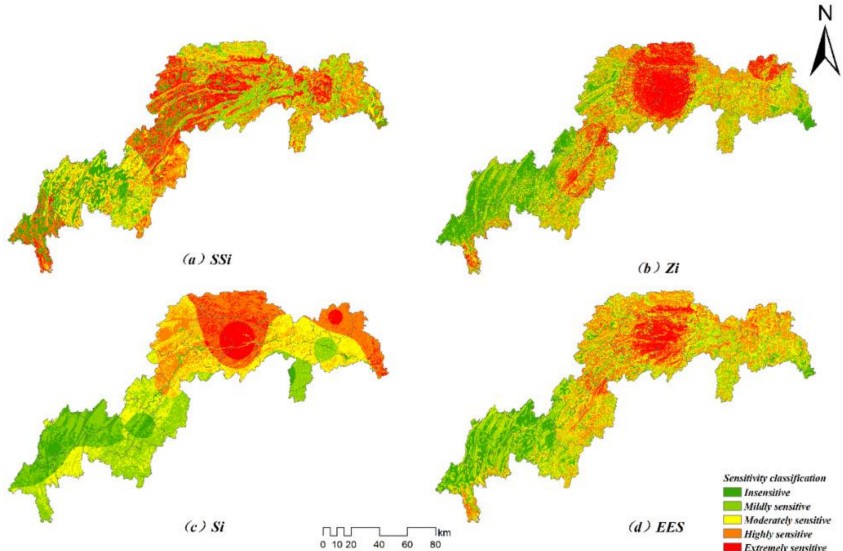

**Figure 4.** The spatial distribution of $SS_i$, $Z_i$, $S_i$ and EES: (**a**) soil erosion sensitivity; (**b**) land desertification sensitivity; (**c**) salinization sensitivity; (**d**) comprehensive ecological sensitivity.

### 2.3.2. CA-Markov Model

The CA-Markov model is composed of CA model, Markov chain and multi-criteria evaluation (MCE) [29]. The CA model is a discrete, finite state composition of the meta-cell model. It can simulate complex dynamic systems with spatial-temporal characteristics according to certain local rules [43]. Markov chains create the transfer matrix and probability between land use types for multiple time periods in the past through spatial comparison analysis, which are the basic data for predicting future land use patterns. MCE refers to the selection of expansion factors to construct a land use transition suitability image collection. CA-Markov model can effectively predict future land use dynamics [44]. The prediction process equation of the CA-Markov model is shown below [45]:

$$Ctj + 1 = F[Ctj, N] \tag{5}$$

where $C(t_j)$ and $C(t_{j+1})$ are the states of the cell at time $t_j$ and $t_{j+1}$, respectively; F is the transition rule; N is the domain of the cellular. In this study, elevation, slope, earthquake, landslide, highway, main road, railroad and population density were selected as the driving factors affecting urban expansion. Among them, elevation, slope, earthquake and landslide are negative indicators, and the rest are positive indicators. The suitability evaluation maps for earthquakes, landslides, highways, main roads and railroads were calculated by the kernel density tool, and finally, the normalized driver maps were obtained (Figure 5).

The Kappa coefficient was applied to check the accuracy of the CA-Markov model simulation results [46]. The formula for calculating the Kappa coefficient is shown below:

$$Kappa = \frac{p_a - p_c}{1 - p_c} \tag{6}$$

$$p_a = \frac{s}{n} \tag{7}$$

$$p_c = \frac{a_1 * b_1 + a_2 * b_2}{n * n} \tag{8}$$

where n is the total number of cell sizes in the raster; $a_1$ is the number of cell sizes in the real raster of the urban land; $a_2$ is the number of cell sizes in the non-urban land; $b_1$ is the number of cell sizes in the simulated raster of the urban land; $b_2$ is the number of cell sizes in the non-urban land; s is the number of cell sizes in the real raster and the simulated raster that correspond to each other.

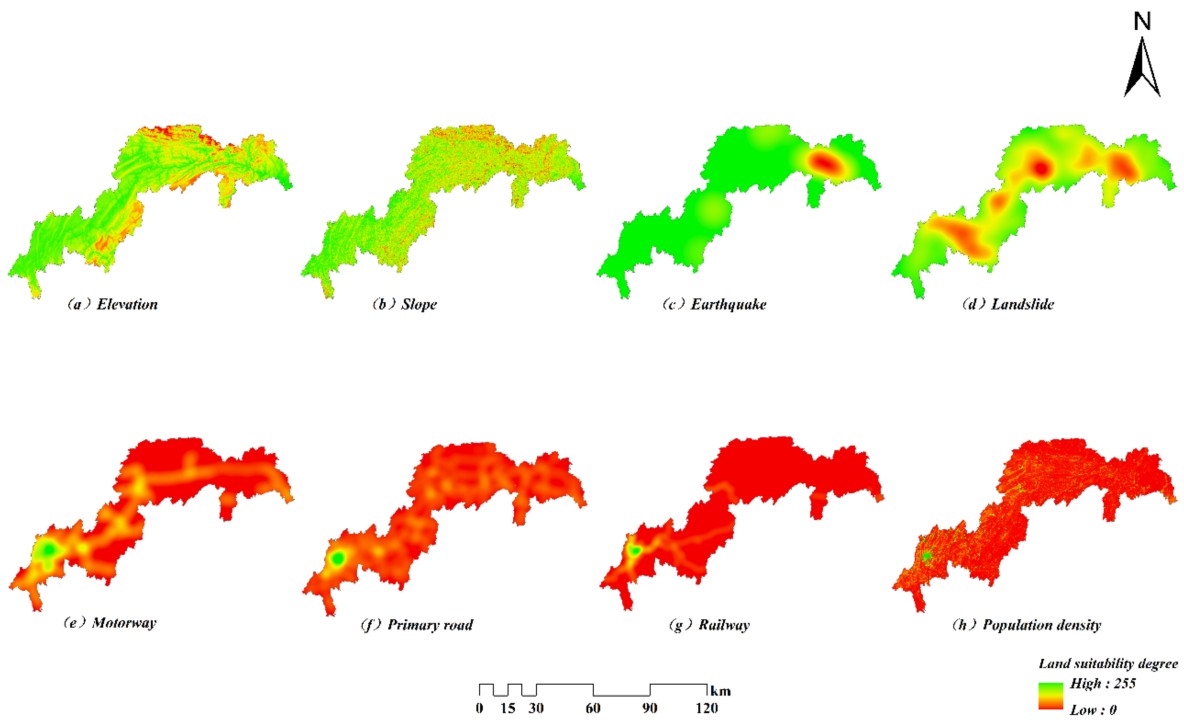

**Figure 5.** The spatial distribution of driving factors of urban expansion in Three Gorges Reservoir area: (**a**) elevation drive factor; (**b**) slope drive factor; (**c**) earthquake drive factor; (**d**) landslide drive factor; (**e**) motorway drive factor; (**f**) primary road drive factor; (**g**) railway drive factor; (**h**) population drive factor.

### 2.3.3. Evaluation of ESV

Ecosystems provide the ecological products and services that humans need [47]. In order to estimate the value of ecosystems, basic value transfer, expert modified value transfer and spatial explicit function modeling were used [5,14,47]. Costanza et al. proposed to divide ecosystem services into 17 services and established a global value equivalence factor system to calculate the global ESV. This method has been widely accepted and used, but it cannot be directly applied to evaluate ESV in China [48]. This paper will be based on the recent research results of Xie et al. in Table 4, classify 6 types of ecosystems into the 4 following types of ecosystem services: provisioning services, regulating services, habitat services and entertainment services. A VPUA-based work has published that in China, and unit equivalent coefficients were estimated as USD 503.2 of ESV per ha [19]. For each ESV category, the calculation formula of $ESV_k$ is as follows:

$$ESV_k = \sum_{i=1}^{n} ESVi \tag{9}$$

$$ESV_k = \sum_{j=1}^{m} EC_{i,j} * \frac{Area_j}{Area_u} \tag{10}$$

where $ESV_k$ is total value of ecosystem services; $ESV_i$ is the ESV for a particular primary service class (i); $EC_{i,j}$ is the equivalent coefficients of the secondary service class (j) in a particular primary service class (i); n is the four kinds of primary service class that include provisioning services, regulating services, habitat services and entertainment services; m is the total number of the secondary service class; $Area_j$ is the area of class (j) in a land type of a hectare and $Area_u$ is a hectare. Table 4 shows the equivalent coefficients table for ESV per unit area in China.

**Table 4.** The equivalent coefficients table for ESV per unit area in China.

| Primary Service | Secondary Service | Farmland | | Forest | | | | Grassland | Water Body | | Unused Land |
| --- | --- | --- | --- | --- | --- | --- | --- | --- | --- | --- | --- |
| | | Dry Land | Paddy Field | Coniferous Forest | Mixed Forest | Broadleaved Forest | Bush | Meadow | Wetland | Lake and River | Barren |
| Provisioning services | Food | 0.85 | 1.36 | 0.22 | 0.31 | 0.29 | 0.19 | 0.22 | 0.51 | 0.80 | 0.00 |
| | Materials | 0.40 | 0.09 | 0.52 | 0.71 | 0.66 | 0.43 | 0.33 | 0.50 | 0.23 | 0.00 |
| | Water | 0.02 | −2.63 | 0.27 | 0.37 | 0.34 | 0.22 | 0.18 | 2.59 | 8.29 | 0.00 |
| Regulating services | Air quality regulation | 0.67 | 1.11 | 1.70 | 2.35 | 2.17 | 1.41 | 1.14 | 1.90 | 0.77 | 0.02 |
| | Climate regulation | 0.36 | 0.57 | 5.07 | 7.03 | 6.50 | 4.23 | 3.02 | 3.60 | 2.29 | 0.00 |
| | Waste treatment | 0.10 | 0.17 | 1.49 | 1.99 | 1.93 | 1.28 | 1.00 | 3.60 | 5.55 | 0.10 |
| | Water flow regulation | 0.27 | 2.72 | 3.34 | 3.51 | 4.74 | 3.35 | 2.21 | 24.23 | 102.24 | 0.03 |
| | Erosion prevention | 1.03 | 0.01 | 2.06 | 2.86 | 2.65 | 1.72 | 1.39 | 2.31 | 0.93 | 0.02 |
| | Maintenance of soil fertility | 0.12 | 0.19 | 0.16 | 0.22 | 0.20 | 0.13 | 0.11 | 0.18 | 0.07 | 0.00 |
| Habitat services | | 0.13 | 0.21 | 1.88 | 2.60 | 2.41 | 1.57 | 1.27 | 7.87 | 2.55 | 0.02 |
| Entertainment services | | 0.06 | 0.09 | 0.82 | 1.14 | 1.06 | 0.69 | 0.56 | 4.73 | 1.89 | 0.01 |
| Total | | 4.01 | 3.89 | 17.53 | 23.09 | 22.95 | 15.22 | 11.43 | 52.02 | 125.61 | 0.2 |

## 3. Results

### 3.1. Ecological Sensitivity Zone Identification

3.1.1. The Evaluation of Soil Erosion Sensitivity

Soil erosion is the serious damage to natural resources of water, soil and land productivity [49].The soil erosion sensitivity result was calculated by formula (2), as shown in Table 5 and Figure 4a. It shows a lower sensitivity at both ends and a higher sensitivity in the middle. The percentages of mildly sensitive zones and moderately sensitive zones are 30.00% and 22.99%, respectively. Next, the highly sensitive and insensitive zones accounted for 19.88% and 15.85%, respectively. Finally, the least percentage of the extremely sensitive zone is 11.28%, and it is mainly concentrated in Yunyang, Zigui, Fengjie, Kaizhou and Wanzhou.

**Table 5.** The comprehensive assessment on ecological sensitivity.

| Sensitivity Classification | Soil Erosion | | Land Desertification | | Soil Salinization | | The Comprehensive Evaluation | |
|---|---|---|---|---|---|---|---|---|
| | Area (km$^2$) | Percentage (%) | Area (km$^2$) | Percentage (%) | Area (km$^2$) | Percentage (%) | Area (km$^2$) | Percentage (%) |
| Extremely sensitive | 6494.08 | 11.28 | 10,587.74 | 18.40 | 5145.91 | 8.94 | 6279.62 | 10.91 |
| Highly sensitive | 11,442.09 | 19.88 | 10,246.66 | 17.80 | 16,129.51 | 28.02 | 14,360.09 | 24.95 |
| Moderately sensitive | 13,232.75 | 22.99 | 13,845.73 | 24.06 | 16,224.49 | 28.19 | 17,261.63 | 29.99 |
| Mildly sensitive | 17,265.48 | 30.00 | 13,213.63 | 22.96 | 14,444.79 | 25.10 | 15,234.81 | 26.47 |
| Insensitive | 9120.03 | 15.85 | 9660.65 | 16.79 | 5609.71 | 9.75 | 4418.26 | 7.68 |

3.1.2. The Evaluation of Land Desertification Sensitivity

Land desertification is generally defined as the loss of surface soil due to soil erosion, a reduction of agricultural land use value and ecological degradation [50]. In the spatial distribution of land desertification sensitivity (Figure 4b), extremely and highly sensitive zones occur mainly in Fengjie County, Wuxi County, Yunyang County and the upper part of Xingshan County. From the amount of land desertification sensitivity, as shown in Table 5, the moderately sensitive zone and the mildly sensitive zone accounted for the highest percentages of 24.06% and 22.96%, respectively. The extremely sensitive zone, highly sensitive zone and insensitive zone accounted for 18.40%, 17.80% and 16.79%, respectively.

3.1.3. The Evaluation of Soil Salinization Sensitivity

Soil salinization is a process in which salts from the soil substrate or groundwater rise to the surface, causing salts to accumulate in the surface soil after the water evaporates [51]. The spatial heterogeneity of the distribution of soil salinity sensitivity in the study area is shown in Figure 4c. The characteristics of the spatial distribution showed a gradual decrease from northeast to southwest. The higher sensitivity zones are mainly in Fengjie County, Wuxi County and Xingshan County. In the evaluation table of soil salinity sensitivity (Table 5), the highest percentages of the moderately and highly sensitive zones are 28.19% and 28.02%, respectively. The mildly sensitive and insensitive zones accounted for 25.10% and 9.75%, respectively. The least percentage was observed in the extremely sensitive zone at 8.94%.

3.1.4. The Comprehensive Evaluation and Zone Identification of Ecological Sensitivity

The comprehensive sensitivity of the ecological environment was calculated by formula (1), as shown in Table 5 and Figure 4d. In general, the highly sensitive zone is mainly concentrated in Wuxi, Yunyang and Fengjie. It is close to the Daba Mountain Nature Reserve and belongs to high forest cover areas with high terrain elevation [52]. The moderately sensitive zone is mainly concentrated in Zhong, Fengdu, Badong and Zigui.

The following zone of low sensitivity is mainly concentrated in the nine central urban areas of Chongqing, Fuling and Jiangjin. As shown in the comprehensive sensitivity evaluation table of the ecological environment (Table 5), the study area mainly showed moderate sensitivity (about 29.99%). The proportions of extremely sensitive, highly sensitive, low sensitive and insensitive zone are 10.91%, 24.95%, 26.47% and 7.68%, respectively.

In the identification of ecological sensitive zone, we use the spatial analyst tool in ArcGIS to extract the extremely and highly sensitive zone in the comprehensive ecological sensitivity, the vector data of Chinese nature reserves were converted into raster grid data, and then both were mosaicked. Finally, we obtained the spatial distribution map of the ecological key zone and the ecological core zone in the Three Gorges Reservoir area (Figure 6).

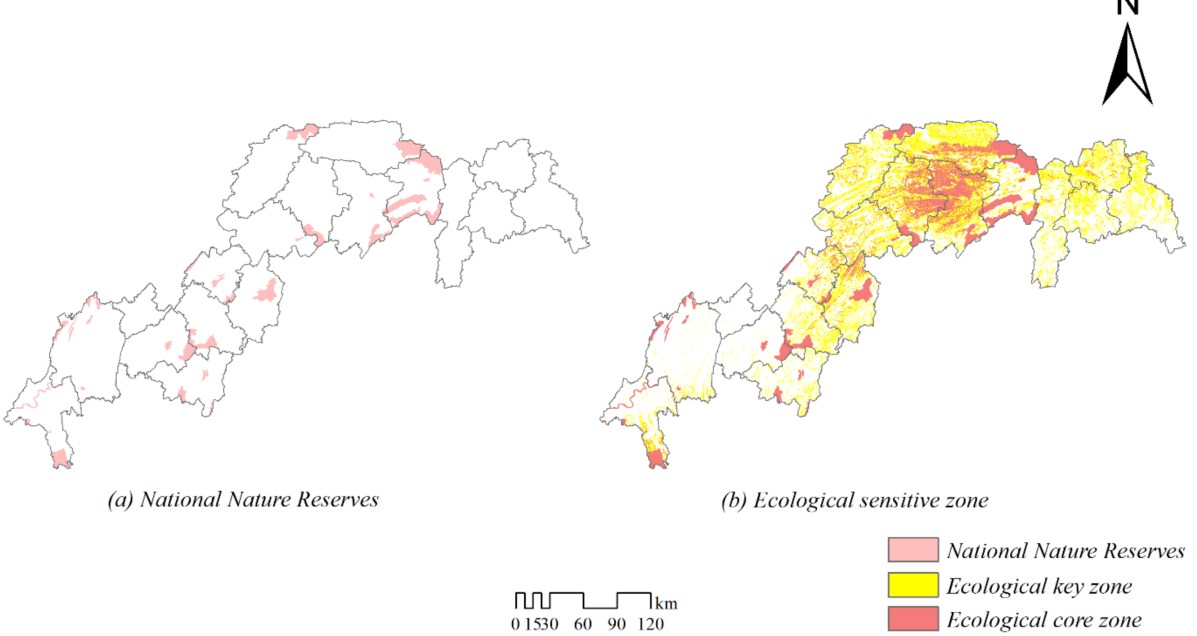

*(a) National Nature Reserves*        *(b) Ecological sensitive zone*

0 15 30   60   90   120 km

National Nature Reserves
Ecological key zone
Ecological core zone

**Figure 6.** The spatial distribution of ecological key zone and ecological core zone.

*3.2. The Urban Expansion Simulation*

3.2.1. The Model Validation and Assessment

Using the 2000 and 2010 actual map as the base data and the 2010 land use data as the initial state and driving factors of urban expansion, we predicted the 2018 urban development pattern (Figure 7d). The accuracy of the model was verified by using the 2018 actual pattern with the 2018 simulated pattern to judge the accuracy of the model. By comparing with the actual 2018 urban pattern (Figure 7c), using the Kappa coefficient formula to verify the accuracy, the Kappa coefficients for the three land types and the whole were calculated in Figure 8. The Kappa coefficients for nonurban, urban and water bodies are 86.73%, 83.14% and 92.19%, respectively, while the accuracy of the overall pattern was 86.74%, which indicated that the overall simulation accuracy was better [53], it can provide a basis for further simulation.

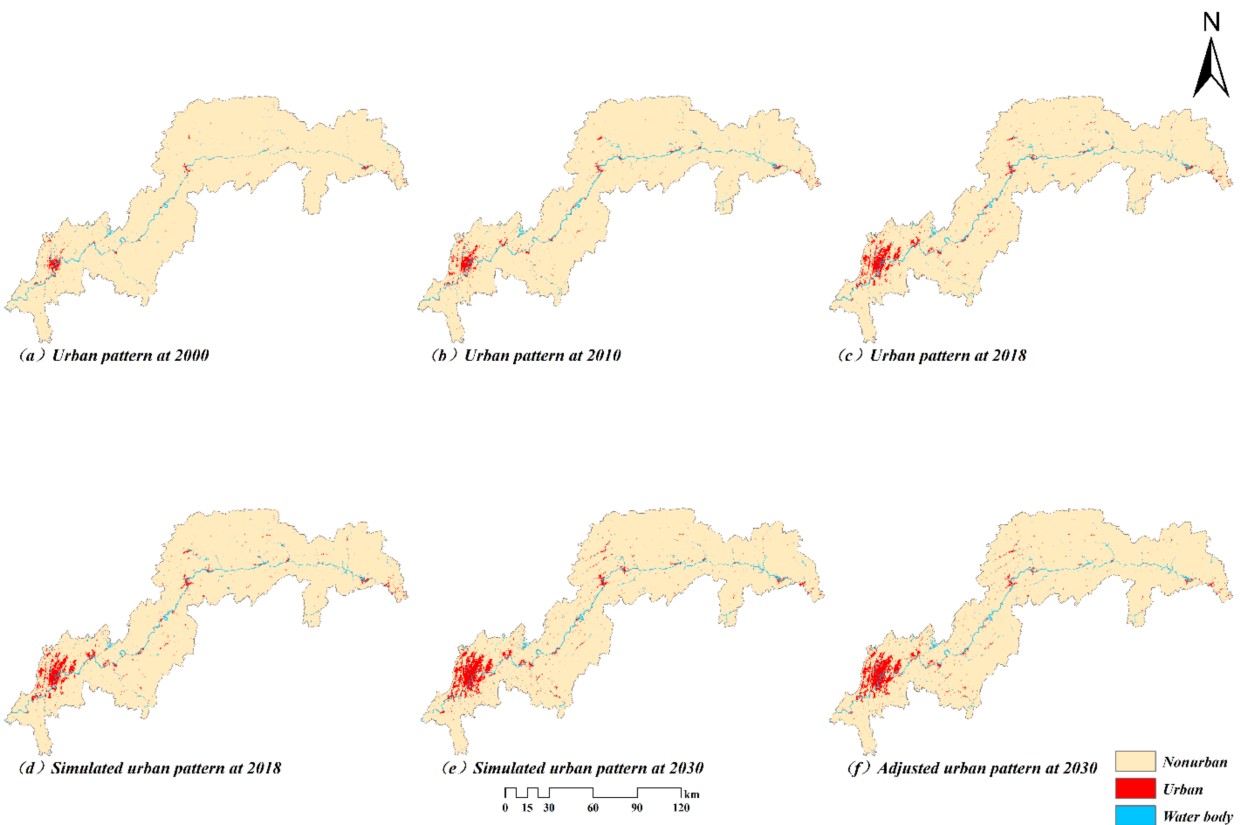

**Figure 7.** Urban patterns for 2000, 2010 and 2018, simulated scenarios for 2018, 2030 and adjusted scenario for 2030 in three classes: (**a**) urban pattern in 2000; (**b**) urban pattern in 2010; (**c**) urban pattern in 2018; (**d**) simulated urban pattern in 2018; (**e**) simulated urban pattern in 2030; (**f**) adjusted urban pattern in 2030.

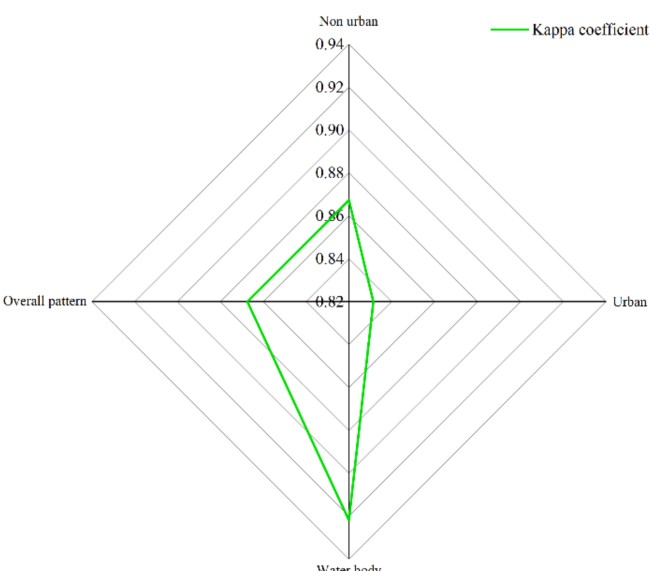

**Figure 8.** Kappa coefficient of three classes and overall pattern.

### 3.2.2. The Simulation of Urban Expansion Based on Ecological Sensitivity

This study will use the trained sample to predict the state of urban expansion in the Three Gorges Reservoir area in 2030, the urban expansion simulation map of the study area in 2030 (Figure 7e) and the area of urban land is about 226,594.43 ha. The focus is on the protection of ecological core zone and ecological key zone in the context of "Ecological

Priority and Green Development" so that the urban expansion can be effectively adjusted. Using spatial overlay and analysis tools, the adjusted land use is shown in Figure 7f. Excluding the part of urban land that overlaps with the ecological core zone and ecological key zone, the area of this part of land is about 15,181.92 ha, and finally, the area of urban land is adjusted to 211,412.51 ha in 2030.

### 3.2.3. The Analysis of the Urban Expansion Simulation

The results of the 2030 urban land expansion were simulated based on ecological sensitivity in the Three Gorges Reservoir area, and we identified the development direction and area change of urban expansion in different districts in the study area. A map of the urban expansion change in the study area from 2018–2030 is shown in Figure 9, the faster growth in urban land is mainly occurring around the main urban areas of Chongqing. The statistical analysis yields a statistical map of the urban expansion area as shown in Figure 10, the northwestern part of Banan shows the most growth in urban land, about 9903.69 ha; followed by the northwestern part of Yubei, with an increase of 9400.41 ha; the growth area of Jiulongpo, Shapingba and Beibei are about 7448.76 ha, 7308.09 ha and 6133.68 ha, respectively; Yuzhong growth area is only 100.80 ha in urban land, and it is the only city-wide urban functional core area in Chongqing and past large-scale urban expansion has resulted in less available land for development and construction. Next, the smallest areas of urban land growth are in Xingshan, Zigui and Wushan. These areas are constrained by natural conditions, economic development and policy regulation, making them unsuitable for future urban expansion.

### 3.3. Estimation of ESV Change

#### 3.3.1. ESV Change from 2000 to 2018

The ESV was calculated by Formulas (9) and (10) and land use data as shown in Table 6, Figures 11 and 12.

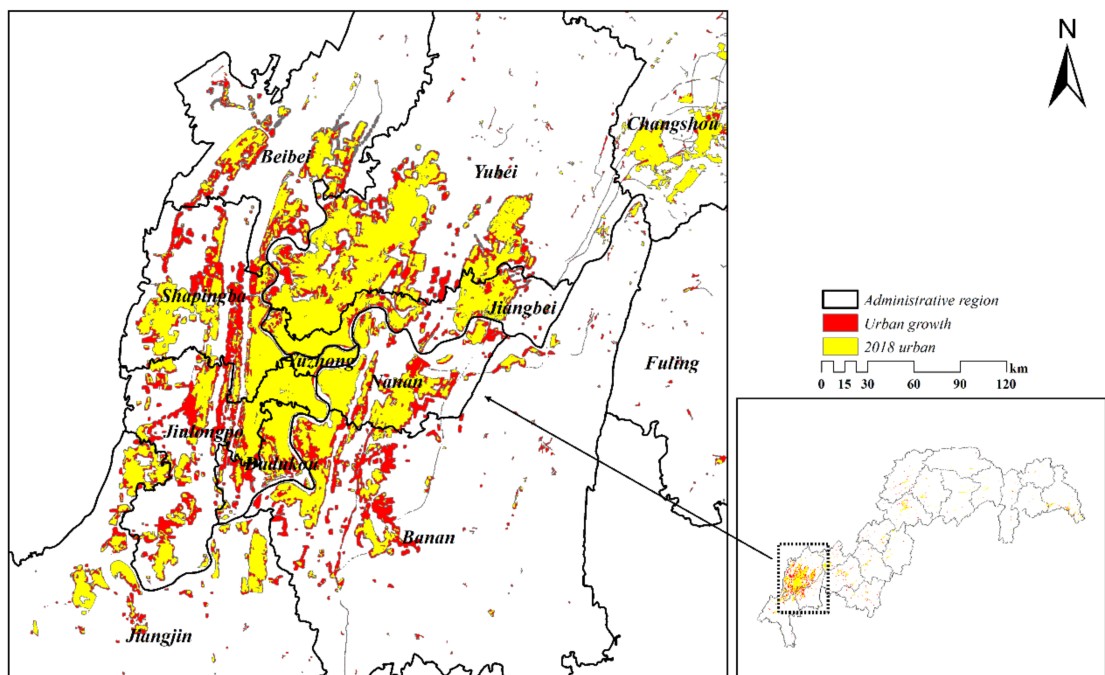

**Figure 9.** Modeled urban expansion from 2018 to 2030.

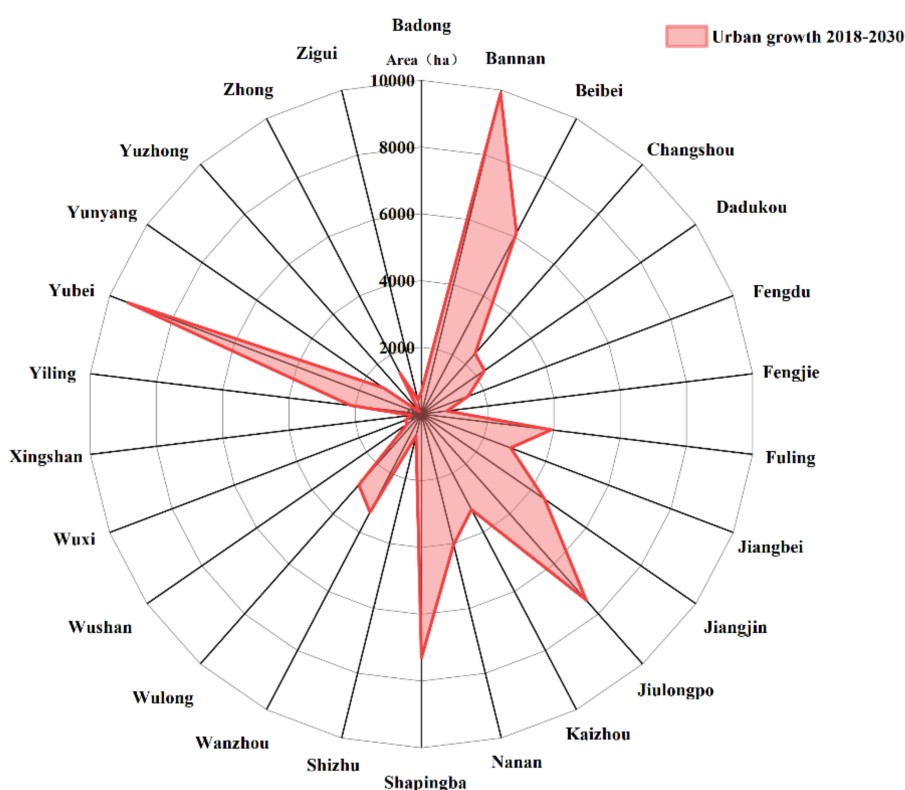

**Figure 10.** The urban growth area of various districts from 2018 to 2030 in the Three Gorges Reservoir area.

**Table 6.** The ESV of different land use types in the study area from 2000 to 2018.

| Land Use Type | 2000 | | 2010 | | 2018 | |
|---|---|---|---|---|---|---|
| | ESV (Million USD) | Percentage (%) | ESV (Million USD) | Percentage (%) | ESV (Million USD) | Percentage (%) |
| Dry land | 3188.74 | 8.32 | 3135.26 | 7.75 | 3105.17 | 7.40 |
| Paddy field | 1229.56 | 3.21 | 1196.58 | 2.96 | 1139.29 | 2.71 |
| Coniferous forest | 7960.03 | 20.77 | 7959.81 | 19.68 | 7171.30 | 17.09 |
| Mixed forest | 719.17 | 1.88 | 883.25 | 2.18 | 876.58 | 2.09 |
| Broadleaved forest | 10,893.48 | 28.43 | 11,314.77 | 27.97 | 16,575.17 | 39.50 |
| Bush | 5923.07 | 15.46 | 6150.63 | 15.20 | 3372.12 | 8.04 |
| Meadow | 4252.15 | 11.10 | 3578.59 | 8.85 | 3413.38 | 8.13 |
| Wetland | 537.39 | 1.40 | 459.82 | 1.14 | 515.56 | 1.23 |
| Lake and river | 3616.79 | 9.44 | 5775.83 | 14.28 | 5796.12 | 13.81 |
| Barren | 0.10 | 0.00 | 0.05 | 0.00 | 0.05 | 0.00 |
| Total | 38,320.49 | — | 40,454.61 | — | 41,964.75 | — |

From Table 6, it can be found that the total ESV of the study area is USD 38.32 billion (2000), USD 40.45 billion (2010) and USD 41.96 billion (2018), respectively. In 18 years, the total ESV of the study area increased by USD 3644.26 million. The ecosystem services of the broadleaved forests, the coniferous forests, the bush, the meadows, the lakes and the rivers provide the main systems in various ecosystem services, and their value accounted for 85.20% (2000), 85.98% (2010) and 86.57% (2018) of the total value. The broadleaved forests had the highest ESV, about 1 USD 0.89 billion (2000), USD 11.31 billion (2010) and USD 16.58 billion (2018); the coniferous forests have the second highest value, about USD

7.96 billion (2000), USD 7.96 billion (2010) and USD 7.17 billion (2018), respectively. From 2000 to 2018, the value of broadleaved forests has increased most significantly, while the value of the dry land and paddy fields has been decreasing. The results indicate that although the Three Gorges Reservoir area has been affected by urban expansion to some extent, the overall value has maintained an increasing trend, mainly due to the increasing area of broadleaved forests, which provide a large amount of ESV.

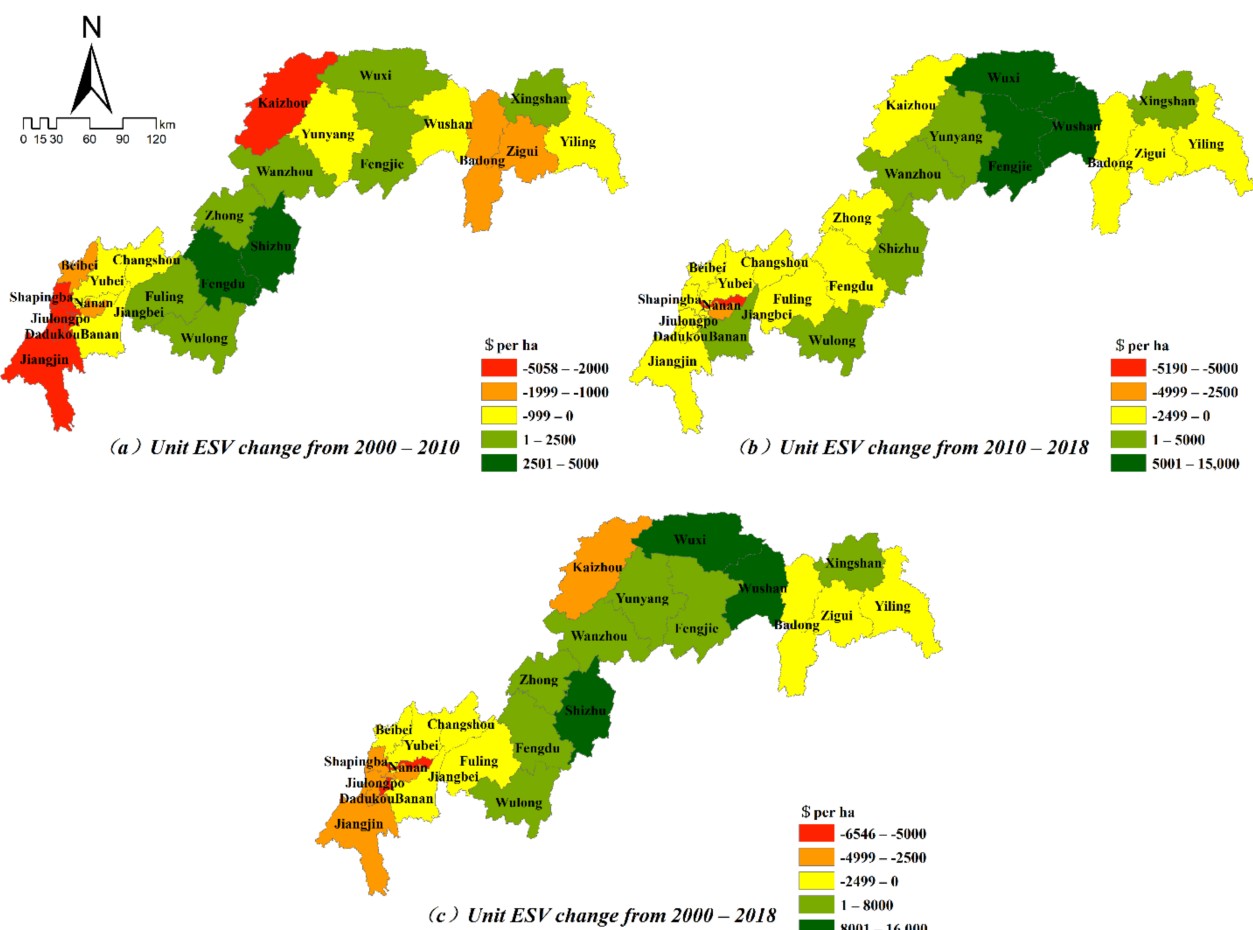

**Figure 11.** The change in unit ESV from 2000 to 2018.

At the district scale, nine districts and counties had increasing unit ESVs from 2000 to 2010 (Figure 11a), this growth is partially concentrated in the middle section of the Three Gorges Reservoir area, with the highest gain in Shizhu (4052.54 USD/ha). However, the decrease in unit ESV is mainly concentrated in the vicinity of the main city of Chongqing: Dadukou had the largest decrease, about 5058.49 USD/ha. The total value showed an increasing trend in the Three Gorges Reservoir area between 2010 and 2018, but there are still 17 districts that declined in unit ESV (Figure 11b), and the most significant decrease was found in Jiangbei (5190.16 USD/ha). The increase was concentrated in the key protective areas of the Three Gorges Reservoir area, such as Wushan and Wuxi, and the highest increase was found in Wuxi County (14,530.61 USD/ha). Overall, the increase in unit ESV during the 18 years was concentrated in the central part of the Three Gorges Reservoir area (Figure 11c), with Wuxi, Wushan and Shizhu growing faster than 8000 USD/ha, and Dadukou and Jiangbei showing the most significant decrease, both exceeding 5000 USD/ha.

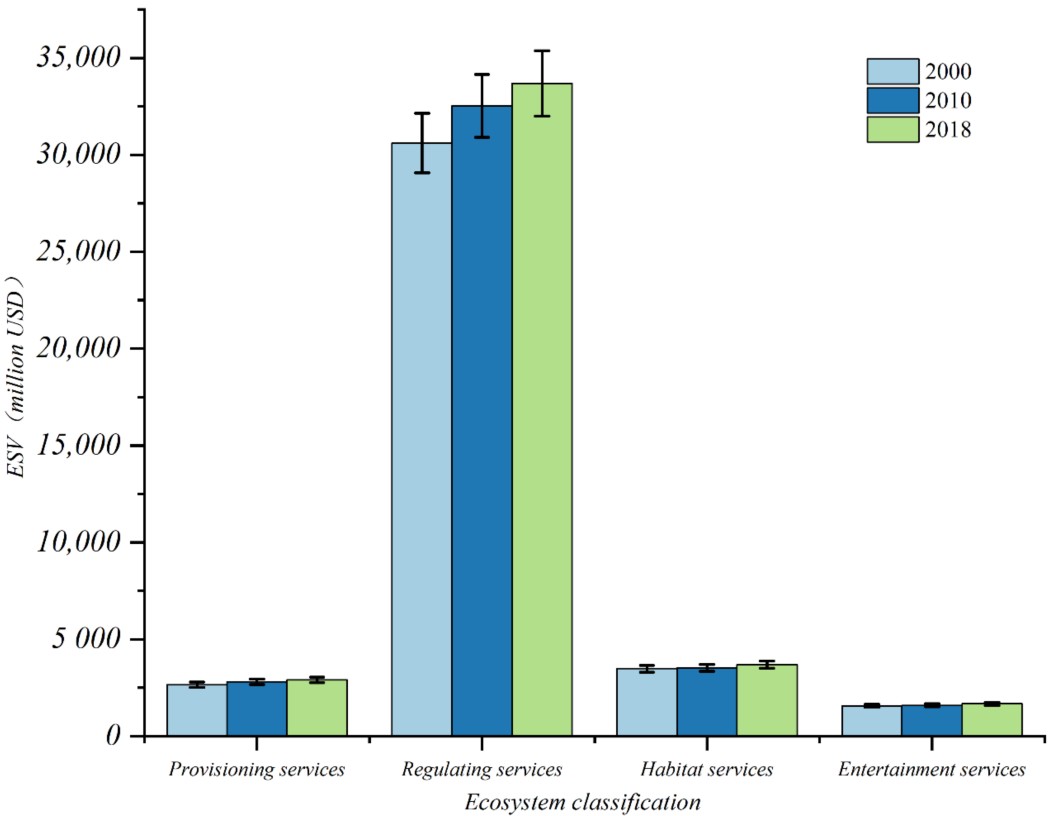

**Figure 12.** The ESV of four ecosystem classifications from 2000 to 2018.

As shown in Figure 12, the four types of value in the study area showed an increasing trend from 2000 to 2018. We found that regulating services accounted for the highest proportion of overall ESV, about 79.88% (2000), 80.40% (2010) and 80.29% (2018), respectively. The proportion of each type of ecosystem service function had the following order of magnitude: regulating services > habitat services > provisioning services > cultural and amenity services.

3.3.2. Prediction of ESV Losses from 2018 to 2030

The CA-Markov model predicts the urban expansion pattern of the study area in 2030, and the results show that there will be a certain degree of outward urban expansion, mainly in the vicinity of the main city of Chongqing. The urban expansion will lead to a decrease in the value of ecosystem services. Our model suggests that the newer urban area is 80,026.02 ha from 2018 to 2030. In the ten land use types (Figure 13a), The most transferred land to urban areas is dry land and paddy field, about 33,903.54 ha and 28,381.23 ha, respectively, while the least transferred area is wetland, lake and river, about 6.48 ha and 61.38 ha, respectively.

Table 7 shows the significant differences in the losses of other ecological land under urban expansion in different districts from 2018 to 2030. Dry land losses in Banan and Yubei comprised an area of 5083.38 ha and 4681.71 ha, respectively, the sum of which accounted for 28.80% of the total area. The area of paddy field losses in Yubei, Banan, Shapingba and Jiulongpo accounted for 45.69% of the total area. The largest coniferous and mixed forest area losses were in Wulong, Beibei and Kaizhou with areas of about 293.22 ha and 268.65 ha lost, respectively. The losses of broadleaved forest mainly occurred in Jiulongpo (1177.83 ha) and Shapingba (1130.31 ha). The losses of meadows mainly occurred in Fuling (509.67 ha). The largest losses occurred in Zhong (4.05 ha) and Beibei (45.90 ha), respectively. In barren land, Jiangbei and Banan lost the largest area of 376.65 ha and 285.48 ha, respectively.

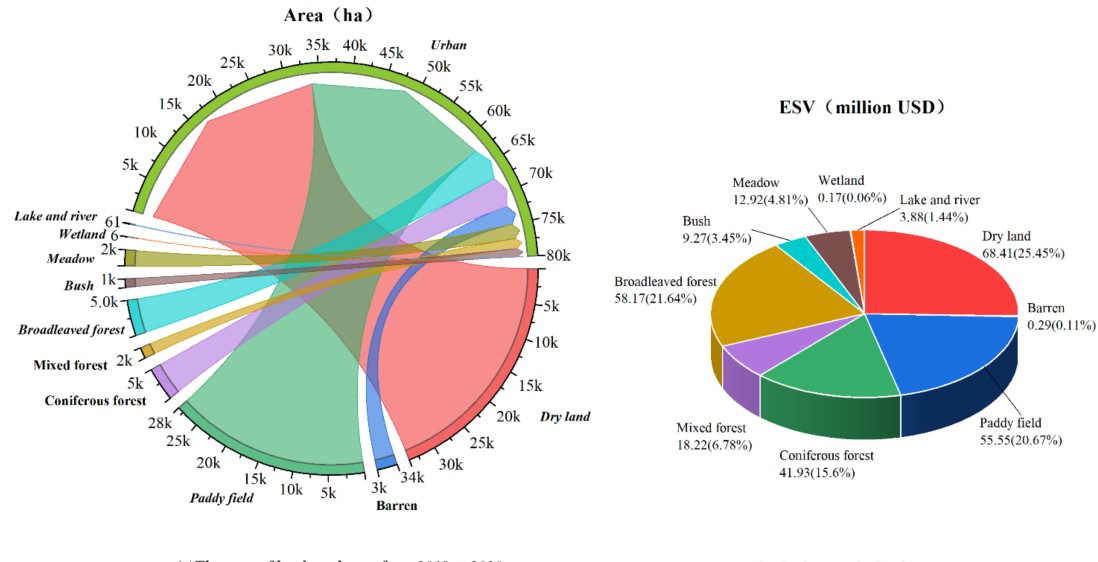

**Figure 13.** The losses area of land-use area and ESV from 2018 to 2030.

**Table 7.** The land-use losses in different districts from 2018–2030.

| Region (ha) | Farmland | | Forest | | | | Grassland | Water Body | Unused Land |
| | Dry Land | Paddy Field | Coniferous Forest | Mixed Forest | Broadleaved Forest | Bush | Meadow | Wetland | Lake and River | Barren |
|---|---|---|---|---|---|---|---|---|---|---|
| Badong | 138.6 | 143.19 | 216.81 | 76.05 | 54.36 | 1.89 | 6.21 | 0 | 0 | 43.2 |
| Xingshan | 47.79 | 61.29 | 42.93 | 73.89 | 15.66 | 13.95 | 1.26 | 0 | 0 | 51.12 |
| Yiling | 209.88 | 767.88 | 318.69 | 71.82 | 402.21 | 271.26 | 4.14 | 0 | 0 | 95.58 |
| Zigui | 22.41 | 108.09 | 201.87 | 13.23 | 45.72 | 21.15 | 2.79 | 0 | 0 | 24.75 |
| Banan | 5083.38 | 3623.40 | 302.04 | 83.79 | 434.88 | 29.25 | 55.44 | 0 | 6.03 | 285.48 |
| Beibei | 3111.93 | 2213.37 | 113.49 | 293.22 | 139.5 | 64.17 | 0.18 | 0 | 45.9 | 151.92 |
| Changshou | 679.77 | 1546.38 | 45 | 0 | 87.12 | 20.07 | 27.09 | 0.99 | 0 | 18.54 |
| Dadukou | 1135.71 | 679.86 | 175.32 | 26.64 | 232.47 | 0 | 0 | 0 | 0 | 48.87 |
| Fuling | 1281.42 | 1466.01 | 337.32 | 14.4 | 153.36 | 1.71 | 509.67 | 0 | 0 | 160.47 |
| Jiangbei | 1221.75 | 840.24 | 168.93 | 214.2 | 38.43 | 0 | 0 | 0 | 0 | 376.65 |
| Jiangjin | 2434.32 | 986.67 | 321.3 | 57.42 | 134.91 | 304.02 | 176.49 | 0 | 0 | 73.8 |
| Jiulongpo | 3026.34 | 2533.14 | 327.24 | 116.19 | 1177.83 | 152.01 | 49.95 | 0 | 0 | 66.06 |
| Nanan | 1906.20 | 1587.96 | 227.7 | 8.64 | 125.19 | 9.36 | 0 | 0 | 0 | 171 |
| Shapingba | 2778.84 | 2978.82 | 225.63 | 1.35 | 1130.31 | 8.01 | 6.3 | 0 | 0 | 178.83 |
| Wanzhou | 1035.72 | 1462.14 | 191.88 | 109.26 | 14.31 | 34.56 | 313.65 | 0 | 0 | 161.91 |
| Yubei | 4681.71 | 3830.94 | 282.87 | 49.05 | 347.04 | 11.25 | 64.8 | 1.44 | 9.45 | 121.86 |
| Yuzhong | 0 | 0 | 0 | 9.18 | 0 | 0 | 0 | 0 | 0 | 91.62 |
| Fengdu | 663.21 | 533.97 | 77.49 | 6.93 | 59.67 | 39.87 | 57.78 | 0 | 0 | 58.68 |
| Fengjie | 196.56 | 235.71 | 86.85 | 0.54 | 51.93 | 8.19 | 96.03 | 0 | 0 | 98.91 |
| Kaizhou | 1342.89 | 1261.80 | 78.3 | 268.65 | 10.44 | 1.98 | 203.85 | 0 | 0 | 62.28 |
| Shizhu | 234.36 | 266.4 | 40.05 | 0 | 20.43 | 3.24 | 73.35 | 0 | 0 | 30.87 |
| Wushan | 231.75 | 78.03 | 4.23 | 1.89 | 119.43 | 5.58 | 16.29 | 0 | 0 | 54.18 |
| Wuxi | 232.38 | 101.07 | 13.41 | 0 | 22.5 | 2.34 | 133.83 | 0 | 0 | 14.85 |
| Wulong | 1284.03 | 434.97 | 484.38 | 0 | 207.63 | 174.96 | 156.69 | 0 | 0 | 95.22 |
| Yunyang | 377.01 | 245.79 | 193.05 | 14.49 | 11.16 | 16.74 | 250.47 | 0 | 0 | 271.8 |
| Zhong | 545.58 | 394.11 | 276.21 | 57.42 | 0.18 | 15.48 | 39.87 | 4.05 | 0 | 49.86 |
| Total | 33,903.54 | 28,381.23 | 4752.99 | 1568.25 | 5036.67 | 1211.04 | 2246.13 | 6.48 | 61.38 | 2858.31 |

The increasing urbanization will encroach on other ecological lands and directly lead to the losses of ESV by the equivalent coefficients. We found that the total ESV losses is USD 268.81 million in the Three Gorges Reservoir area (Figure 13b). The highest values of dry land and broadleaved forest were lost with USD 68.41 million (25.45%) and USD 58.17 million (21.64%), respectively. The variation of the overall ESV losses was explored from the district scales (Figure 14a). Jiulongpo, Banan, Shapingba and Yubei have higher losses (more than USD 25 million). This result indicates that these areas are at risk of ecological degradation. Yuzhong, Xingshan and Wuxi have the lowest losses, mainly due to the highest level of urbanization in Yuzhong, resulting in limited land for urban development. In addition, Xingshan and Wuxi are as important ecological reserves in China, which limit

the large-scale expansion of urban land use. On average, Dadukou, Jiulongpo, Shapingba and Nanan have the highest unit ESV losses (more than 400 USD/ha) (Figure 14b).

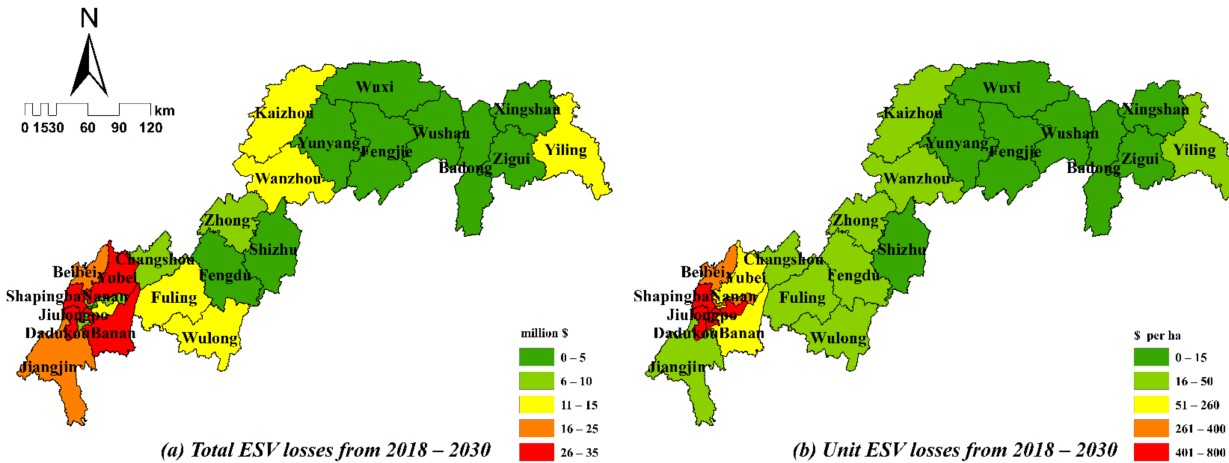

**Figure 14.** The losses in the total and unit ESV from 2018 to 2030.

Figure 15 summarizes the amount of loss of the four primary services in different districts. In Figure 15a, we find that Banan, Jiulongpo and Jiangjin have the largest loss of provisioning services values (more than USD 1.2 million). There are 13 districts and counties with a loss of less than USD 0.3 million, mainly concentrated in the upper half of the Three Gorges Reservoir area of Wushan, Badong and Zigui, etc. The losses of regulation services values are the largest among the four types of services. Four districts lost more than USD 20 million: Jiulongpo, Banan, Shapingba and Yubei (Figure 15b). These districts are important places for future urban expansion and their ecological regulating services functions will decline from 2018 to 2030. Among the habitat services values and entertainment services values (Figure 15c,d), Jiulongpo and Shapingba have the largest losses.

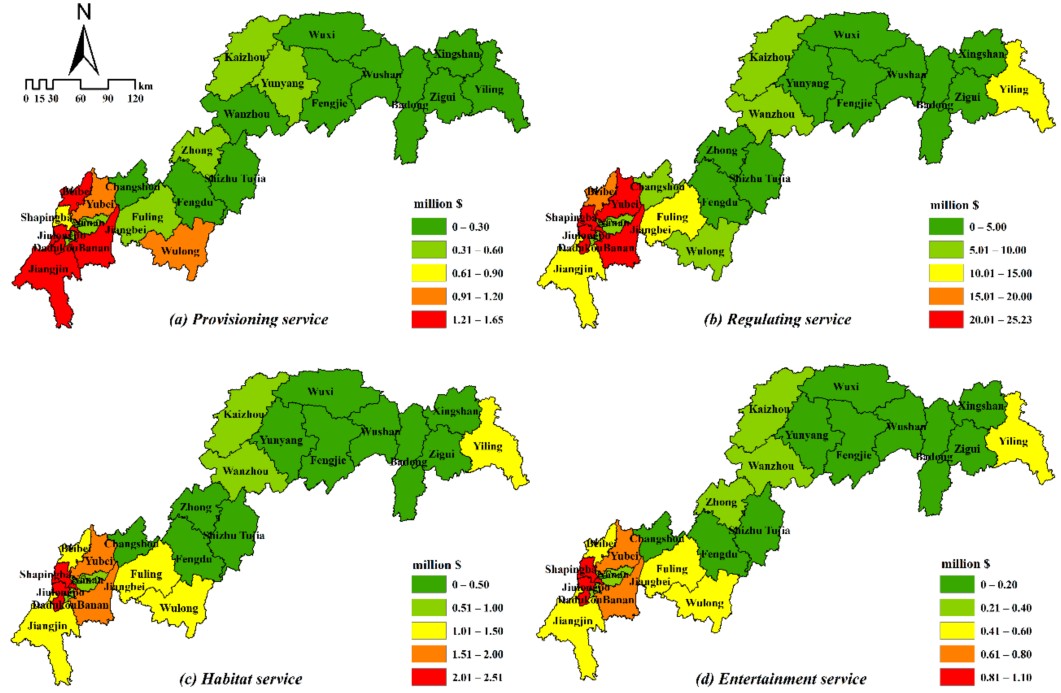

**Figure 15.** The ESV losses in primary services from 2018 to 2030.

## 4. Discussion

### 4.1. The Rationality of Delimiting Ecological Sensitive Areas

First, because of the limitation of collecting data and the need to delimit ecological sensitive areas, according to the actual situation of the Three Gorges Reservoir area, this study selects relevant indicators from 2018 to identify the ecological sensitive areas in the study area in terms of three sensitivities (i.e., soil erosion sensitivity, land desertification sensitivity and soil salinity sensitivity) so as to identify the scope and number of ecological sensitive areas and provide a basis for achieving green development in the region [54]. As of now, the indicators and data will be further updated, which is a hot topic for future research.

Second, the grid size used to identify ecological sensitive areas is 30 m × 30 m in this paper. However, some of the raw data (NDVI, soil type, groundwater mineralization and groundwater depth) are of higher resolution, but still can accurately identify the extent of ecological sensitive zone. We believe that higher resolution data will be used in the future to improve the accuracy of delimiting ecological sensitive zones.

Third, in the comprehensive ecological sensitivity evaluation, the areas with high sensitivity or above account for 35.86% of the study area, with an area of 20,639.71 km$^2$, mainly in Wushan, Fengjie and Yunyang. This part is a forest reserve with mainly mountainous terrain and high elevation, and its ecosystem types are diversified and spatially heterogeneous, which are vulnerable to some natural disasters and human factors. The harm to the ecological sensitive zone should be reduced. Meanwhile, a corresponding ecological compensation mechanism is established [55], the ecological advantages will be transformed into economic advantages, the ecological and economic benefits of the study area will be improved, and the green development concept of "making all-out efforts to protect it, and forbidding large-scale development" will be fully implemented.

### 4.2. The Accuracy of Simulated Urban Expansion

This study combines the ecological sensitivity, CA and Markov models in the context of green development to simulate and predict the future urban spatial patterns in the Three Gorges Reservoir area. The urban construction land area is predicted to increase by 80,026.02ha between 2018 and 2030, while the main expansion areas are in the northwestern part of Banan, the northwestern part of Yubei, Jiulongpo, Shapingba and Beibei. The analysis revealed that the future urban expansion in the study area is mainly occupying paddy fields and drylands, occupying an area of 33,903.54ha and 28,381.23ha, respectively.

The driving factors affecting urban expansion were mainly selected as elevation, slope, earthquake, landslide, highway, main road, railroad and population density. These factors performed relatively well in simulating urban expansion in 2018, with an overall kappa coefficient of 86.74% compared to the actual 2018 urban sites, but urban expansion is a complex dynamic system [56]. It is not enough to consider only the above eight driving factors. In the future, more socioeconomic and ecological data should be obtained to provide a basis for improving the accuracy of urban expansion simulation. At the same time, different prediction scenarios [57–59] and different grids sizes [33,60] play different roles in the simulation of urban expansion.

### 4.3. Estimation of ESV Dynamics and Prediction of Future ESV Losses as an Effective Tool for Ecological Safety Management

The estimation of ESV can provide a basis for the region to achieve sustainable development [61]. The ESV change in the study area were calculated by using the land use data from 2000–2018 and the ESV value equivalent table. The results show that the ESV in the Three Gorges Reservoir area increased from USD 38,320.49 million in 2000 to USD 41,964.75 million in 2018, and the overall ESV change showed an increasing trend. This may be due to the continuous increase in the area of forest land (especially broadleaf forest) due to the implementation of the reforestation project, which also brings an increase in the value of different ecosystem service functions [62,63]. However, it is noteworthy

that between 2000 and 2018, there are 16 districts and counties with decreasing ESV per unit. The most significant decreases (more than 5000 USD/ha) were observed in Dadukou and Jiangbei. With increasing urbanization, we see that ESV will show a decreasing trend and ecosystems will face increasing pressure in the region. In the four different ecosystem services, regulating services provide the highest value, followed by habitat services, and finally provisioning services and entertainment services.

As urbanization continues, it may cause degradation of the ecosystem [64,65]. This study uses the CA-Markov model to predict the trend of urban expansion in 2030, and the study finds that the growth of urban land in the study area is mainly through the occupation of dry land and paddy fields. From 2018 to 2030, the overall ESV of the Three Gorges Reservoir area will be reduced by about USD 268.81 million. At the district and county scales, the ESV of Jiulongpo, Banan and Shapingba will be reduced most significantly, and the ecological pressure in the area will be further increased. In starting from the perspective of the four ESV, we found that regulating services values decreased the most. Therefore, it is necessary to combine the ecological sensitivity method and ESV evaluation results to achieve a more reasonable and targeted implementation of ecological spatial management and steadily realize the green development strategy.

This study provides insights into ecological conservation under sustainable urbanization by predicting change in urban land expansion and calculating ESV, using the Three Gorges Reservoir area as an example, which can be of guidance to other countries. Especially, developing countries are facing different problems caused by rapid urbanization, such as India [66], Sri Lanka [53] and Egypt [67]. Each country faces different ecological problems. We should consider geohazard sensitivity and select suitable indicators for the region where geohazards occur frequently. In ESV evaluation, there are differences in different countries' ecosystem value, and the equivalent coefficients table should be changed appropriately according to the regional characteristics. Although there are differences between the spatial scales of different countries, it is necessary to identify the ecological sensitive zones and to predict the future development direction of urban land and the dynamic change of ESV through scientific and rational identification. This quantitative evaluation provides a reference basis for ecological spatial control and effective macro-control.

## 5. Conclusions and Outlook

In the context of economic globalization, the urbanization process of human beings has become irreversible. Handling the relationship between environmental protection and economic development has become particularly critical. In this study, we simulate the future urban expansion and ESV change in the study area based on ecological sensitivity. Under the guidance of green development concept, the socioeconomic maximization is achieved, and the relationship between ecological protection and urban expansion is coordinated. This study shows that:

1. In the comprehensive ecological sensitivity assessment, we found that the ecological sensitive zone is about 20,639.71 km$^2$ in the Three Gorges Reservoir area, accounting for 35.86% of the total study area. This part of the area is in Wushan, Fengjie and Yunyang.
2. The results of the study show that the overall ESV in the Three Gorges Reservoir area showed an increasing trend from 2000 to 2018. The growth was about USD 3644.26 million. From the perspective of ESV change in districts and counties, we found that 16 districts and counties were decreasing in unit ESV, Dadukou and Jiangbei decreased most significantly. In four ecosystem services, regulating services provided the highest ESV.
3. In the context of ecological priority and green development, the 2030 urban land was predicted and simulated. In 2018–2030, about 80,026.02 ha of new construction land will be added to the Three Gorges Reservoir area, and the overall ESV will lose USD 268.75 million. The largest losses are in Jiulongpo, Banan and Shapingba.

Although, this method can better delimit ecologically sensitive zones and estimate the dynamic changes of ESV in the Three Gorges Reservoir area, there is still much room for improvement. We will use high-resolution land use data and remote sensing inversion data (FVC, WET, NDBSI, LST and NPP) in the future work. This allows us to better identify zones of ecological sensitivity. Meanwhile, we should also make different simulations of urban expansion for different scenarios, such as developmental orientation or ecological orientation. It can provide better prediction and estimation of ESV changes for future urban expansion in the Three Gorges Reservoir area.

**Author Contributions:** Data curation, H.P. and J.L.; Funding acquisition, X.Z.; Investigation, H.P. and J.L.; Methodology, H.P., L.H. and X.Z.; Supervision, X.Z. and X.Y.; Visualization, H.P.; Writing—original draft, H.P.; Writing—review & editing, L.H., X.Y. and J.L. All authors have read and agreed to the published version of the manuscript.

**Funding:** This research was supported by the special scientific research project of Hubei Provincial Land Consolidation Center in 2021 and the special scientific research project of Hubei Research Station for Integrated Land Remediation and Restoration in the Jianghan Plain in 2021.

**Institutional Review Board Statement:** Not applicable.

**Informed Consent Statement:** Not applicable.

**Data Availability Statement:** Not applicable.

**Conflicts of Interest:** The authors declare no conflict of interest.

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
