# Peer review of "Evaluation of ESV Change under Urban Expansion Based on Ecological Sensitivity: A Case Study of Three Gorges Reservoir Area in China"

_sustainability, doi:10.3390/su13158490_

Round 1

Reviewer 1 Report

A. General comments

It is a very valuable paper presenting a lot of methodical, as well as factual information.

My basic theoretical-methodical coments concerns 2 areas:

  1. The article is focused on ESV, but there is no explanation what authors consider under the term ecosystem. There is still valid the classic definition originated actually by Haeckel, that ecosystem is the system of biota and its abiotic surroundings (in different verbal primpings). At least, it is needed to state, that the authors consider as „ecosystems“  merely the very much simplified classes of land-use types.                    Tha lack of complex consideration of the concept of ecosystem resulted that the part of the evaluation of sensitivity is based prevailing on the abiotic factors (+ very simplified Ci and Ei factors), and, in opposite, the part of the ESV evaluation exclusivelly on the simplified classes of land-use.
  2. The process and formulas of all individual sensitivity and ESV evaluations themselves can be considered as correct. Nevertheless, there is not vissible connection betveen these 2 part. The seriously performed sensitivity evaluation is not transformed to ESV evaluation. The ESV evaluation based on both, the sensitivity values and on the coefficients of ESV would bring more correct and real data. Just consider the simply examples: it is quite obvious, that all ecosystem services - let say of forests- must have  different values on areas with high or low erosion danger, or with low or high salination. But Tab 4. does not differ the values (?)

Of course, I know, that many of current main-stream studies on ESV neglect this base – unfortunatelly. The simplified classes of land-use types are,  of course, also bearers of ecosystems, but the variety of possible combination of seriously evaluated sensitivity factors with land-use types would give surely more dependable results.

Considering that also the final results might change (?)

Please give, your statement to these comments.

B. Topical comments

  1. Row 86 - Better expression as topographic relief increase is probably increase of the elevation above sea level, or relative hights, or dissection of the relief .. Relief is the general expression for the earth surface.
  2. Row 112 – probably the interpunction is missing, or a new item (6) or more should be included. Just as note - NDVI are actually also statistical data calculated as shares and averages.
  3. The whole chapter 2.3.1 is devoted to the sensitivity evaluation. But, just as  a note: The result values can be named as „sensitivity“, but  these evaluations are actually evaluations of the potential towards erosion intensity (based on the classic Wischmaier-Smith model), intensity of desertification, intensity of salination on certain areas. What should be the decisive difference?

  1. To Tab. 1: The SGi factor is presented as verbal characteristic. The values of factor were ordered to 5 classes, but I doubt, that on such a diverse territory only this 5 categorries of soils occured (?) Other doubt: how could the aouthors order the Ri factor, LSi  factor and  Ci factor order only to  these 5 combinations. I think that there is a number if different combinations of these 4 factors.

The above mentioned is valid also for Tab.2  and Tab.3.

  1. To Fig.3

Considering the diversity of the area,  the sensitivities marked as  ( c ) and (d ) look to homogenous and favorit (?)

  1. To Fig. 6. What is the difference between core zone and key zone? Is the key zone the buffer zone for core zone? The key zone occupies quite a big terrotory. And what is the diference between the National Nature Reserves and core zone. They are the same territoties?

Reviewer 2 Report

The article approaches a very interesting and timely topic focused on a methodology for assessing spatio-temporal changes of ESV and predicting the evolution trend under urban expansion pressure. The article perfectly fits with the journal scope and has a clear title, abstract, keywords and structure.

Figures support the study development, and I would reccomend to provide a higher resolution version in order to increase readability.

Specifically, the selection of the Three Gorges Reservoir in China as the study Case is highly appropriate due to the spatial complexity of the area, as well as the urban and environmental context.

Nevertheless, some minor issues have been detected and are put under the authors’ consideration:

  1. Figures writing: I reccomend to use the comma to separate the thousand digits (e.g.: line 14 abstract 3,644.26 instead of 3644.26). This must be consistent throughout the paper. (lines 319,338,362-363,470 , 489) (table 5)
  2. As long as I have understood from the text, the word "delineate" or "delineating" is use as a synonim for "delimit" - Introduction, section 4.1, among others-. I would reccomend to better use "delimit" / "define"/ or, "enclose" because in my opinion are more accurate with the fact of determining the precise boundaries of the sensitive zones, which entails more than than representing a limit.
  3. In the introduction (lines 35 to 38) " to assess ESV for the region by remote sensing", or " to improve ESV in the region" ... Which region? Is it a specific one or is a general statement? If is the first case, the specific name of the Region should be included, meanwhile in the second case it should be better state "in a region".
  4. Some errors are detected regarding the references (line 157, 164,221,232,...)
  5. Results line 515 this study "can be the guidance to other developing countries"... From the methodological viewpoint the study should be appropriate for any territories, underdevelopment or developed. My question is if there are any specific limitations for appliying this methodology that should be note and include as part of the discussion.
  6. It would be great to include the authors' suggestions for future work in the conclusions section
  7. Minor changes are required and figures should be at a higher quality. 

I would like to congratulate the authors for this work.
